# *R-Zero*: SELF-EVOLVING REASONING LLM FROM ZERO DATA

**Chengsong Huang**[1,2†], **Wenhao Yu**[1†], **Xiaoyang Wang**[1], **Hongming Zhang**[1], **Zongxia Li**[1,3], **Ruosen Li**[1,4], **Jiaxin Huang**[2], **Haitao Mi**[1], **Dong Yu**[1]
[1]Tencent AI Seattle Lab, [2]Washington University in St. Louis,
[3]University of Maryland, College Park, [4]The University of Texas at Dallas
† Core contributors
chengsong@wustl.edu; wenhaowyu@global.tencent.com

## ABSTRACT

Self-evolving Large Language Models (LLMs) offer a scalable path toward super-intelligence by autonomously generating, refining, and learning from their own experiences. However, existing methods for training such models still rely heavily on vast human curated tasks and labels, typically via fine-tuning or reinforcement learning, which poses a fundamental bottleneck to advancing AI systems toward capabilities beyond human intelligence. To overcome this limitation, we introduce *R-Zero*, a fully autonomous framework that generates its own training data from scratch. Starting from a single base LLM, *R-Zero* initializes two independent models with distinct roles – a **Challenger** and a **Solver**. These models are optimized **separately** and **co-evolve** through interaction: the Challenger is rewarded for proposing tasks near the edge of the Solver's capability, and the Solver is rewarded for solving increasingly challenging tasks posed by the Challenger. This process yields a targeted, self-improving curriculum without any pre-existing tasks and labels. Empirically, *R-Zero* substantially improves reasoning capability across different backbone LLMs, e.g., boosting the Qwen3-4B-Base by +6.49 on math reasoning benchmarks, and +7.54 on general-domain reasoning benchmarks.

Code: https://github.com/Chengsong-Huang/R-Zero.

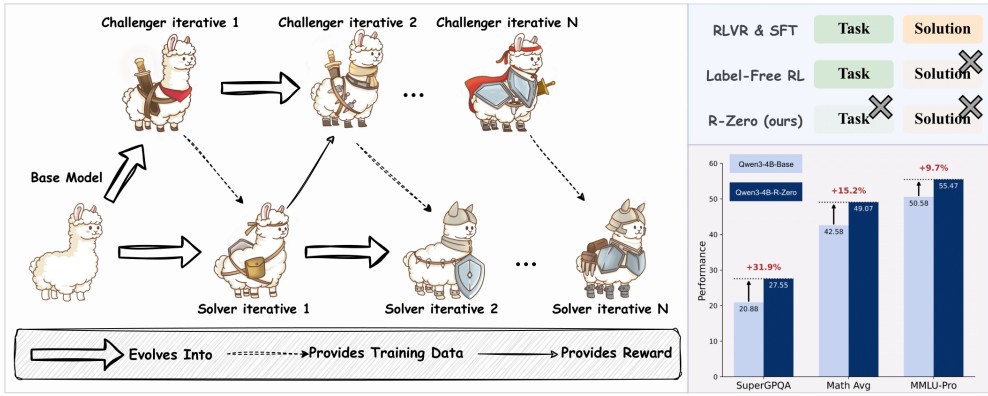

Figure 1: (Left): *R-Zero* employs a co-evolutionary loop between Challenger and Solver. (Right): *R-Zero* achieves strong benchmark gains without any pre-existing tasks or human labels.

## 1 INTRODUCTION

Self-evolving Large Language Models (LLMs) represent a promising frontier for advancing language intelligence. By autonomously generating, refining, and learning from their own experiences, these models provide a scalable pathway toward artificial superintelligence (Tao et al., 2024; Tan

et al., 2024). A critical requirement for training such self-evolving LLMs is access to large volumes of expertly curated tasks and labels, which serve as supervision signals for fine-tuning or reinforcement learning with verifiable rewards (RLVR) (Shao et al., 2024; DeepSeek-AI et al., 2025). However, relying on human annotators to create these tasks and labels is not only costly, labor-intensive, and difficult to scale, but also presents a fundamental bottleneck to advancing AI toward capabilities that could eventually surpass human intelligence (Su et al., 2025; Zhao et al., 2025a).

To reduce dependence on human-curated data, self-generated and label-free methods have been proposed to eliminate the need for explicit supervision. In particular, label-free RL derives reward signals directly from the model's own outputs, such as sequence-level confidence scores (Li et al., 2025a; Prabhudesai et al., 2025; Huang et al., 2025) and output entropy (Agarwal et al., 2025; Cheng et al., 2025). However, despite removing the need for explicit labels, label-free methods still relies on a pre-existing corpus of tasks, which limits its scalability in truly self-evolving settings. On the other side, self-challenging approaches train LLMs on tasks generated by the models themselves (Zhou et al., 2025a; Wang et al., 2025a; Zhao et al., 2025a). While promising, many of these methods rely on external code executors to ensure that the synthesized tasks are both feasible and verifiable. However, in domains that lack an explicit verification oracle, such as open-ended reasoning, ensuring the quality and correctness of self-generated data remains a significant challenge.

In this paper, we propose *R-Zero*, a framework for training reasoning LLMs that can self-evolve from zero external data. In *R-Zero*, a single base model is initialized with two roles – a **Challenger** and a **Solver** that are independently optimized but **co-evolve** throughout the RL process. During co-evolving, the Challenger is rewarded for generating tasks targeted to be at the edge of Solver's current abilities, while the Solver is rewarded for solving increasingly challenging tasks posed by the Challenger. Framework details are provided in Section 2, but briefly, in the Challenger training phase, the Challenger is trained via Group Relative Policy Optimization (GRPO) (Shao et al., 2024) to generate difficult questions. The reward signal is derived from the uncertainty for the frozen Solver, which is measured by the self-consistency of its multiple generated answers. In the Solver training phase, the Solver is fine-tuned with GRPO on a filtered set of these challenging questions generated by the now-frozen Challenger, using the pseudo-labels voted by itself. This entire process repeats, creating a self-evolving cycle that operates without any human intervention.

Our experiments demonstrate that *R-Zero* is a model-agnostic framework, consistently and iteratively improving the reasoning abilities of different backbone LLMs. For example, Qwen3-4B-Base model's average score on math benchmarks increased by a significant **+6.49** points after three iterations of self-evolution. Moreover, the reasoning skills learned through our math-focused questions can generalize to complex general-domain tasks, with models trained using *R-Zero* showing significant improvements on general domain reasoning benchmarks like MMLU-Pro (Wang et al., 2024) and SuperGPQA (Du et al., 2025). Our further analysis finds that *R-Zero* can act as a mid-training method, as models first improved by our method achieve higher performance after fine-tuned on labeled data. In addition, we provide an in-depth analysis that validates our framework's components, demonstrates its synergy with supervised fine-tuning, and characterizes the co-evolutionary dynamics to identify both strengths and limitations, offering insights for future research.

## 2 METHOD

### 2.1 OVERVIEW

We propose ***R-Zero***, a fully automated framework featuring a **Challenger** and a **Solver**, both initialized from the same base LLM. The framework operates in an iterative loop. We illustrate the main framework in Figure 2. First, the Challenger ($Q_\theta$) is trained with Group Relative Policy Optimization (GRPO) to generate synthetic questions that are challenging for the current Solver (Sec. 2.2). A training dataset of question-answer pairs is then constructed from these synthetic questions using a filtering strategy and a majority-vote mechanism (Sec. 2.3). Next, the Solver ($S_\phi$) is fine-tuned on this new dataset, also using GRPO (Sec. 2.4). This iterative process allows the Challenger and Solver to co-evolve, leading to a progressively more capable Solver. The entire framework is self-supervised, requiring no human intervention.

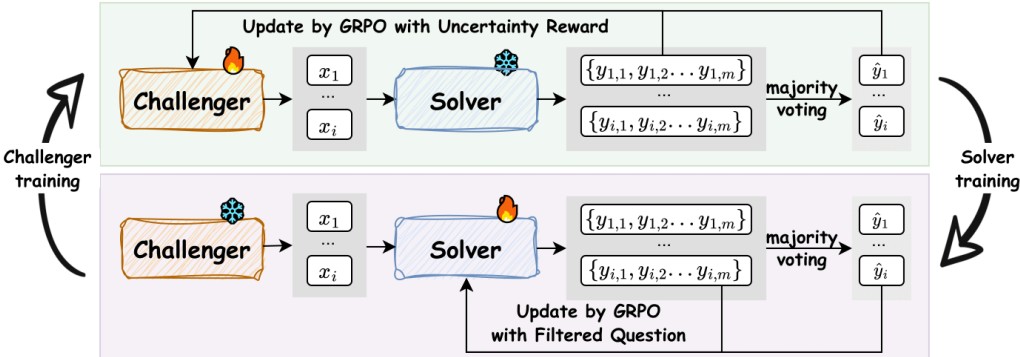

Figure 2: An overview of our *R-Zero* framework, which illustrates the co-evolution of the Challenger and the Solver. **Top:** In the Challenger training phase, the Challenger is trained via GRPO to generate difficult questions. The reward signal is derived from the uncertainty for the frozen Solver, which is measured by the self-consistency of its multiple generated answers. **Bottom:** In the Solver training phase, the Solver is fine-tuned with GRPO on a filtered set of these challenging questions generated by the now-frozen Challenger, using the pseudo-labels voted by itself.

## 2.2 CHALLENGER TRAINING

The Challenger, $Q_\theta$, is an autoregressive language model trained to generate challenging questions. We train $Q_\theta$ using the GRPO algorithm detailed in Sec. I. The core of this process lies in designing a reward function that accurately captures the desired properties of a "good" question. This final scalar reward, $r_i$, is then used in the GRPO. We focus on generating questions specifically within the domain of mathematics, as it provides a convenient and self-contained setting for our framework; the objective nature of mathematical answers allows for the straightforward generation of pseudo-labels via majority voting, without the need for external verification environments like code executors.

**Uncertainty Reward.** To guide the Challenger toward producing challenging yet solvable questions, we first define an uncertainty score. For a generated question $x$, we query the current Solver $S_\phi$ for $m$ responses $\{y_1, \ldots, y_m\}$. The most frequent response is treated as the pseudo-label $\tilde{y}(x)$, and we compute the Solver's empirical accuracy as $\hat{p}(x; S_\phi) = \frac{1}{m} \sum_{j=1}^{m} \mathbb{1}\{y_j = \tilde{y}(x)\}$. The uncertainty reward is then defined as:

$$r_{\text{uncertainty}}(x; \phi) = 1 - 2 \left| \hat{p}(x; S_\phi) - \tfrac{1}{2} \right|$$

This function incentivizes questions where the Solver is maximally uncertain (accuracy approaches 50%). We provide a theoretical motivation for this reward function in Appendix F.

**Repetition Penalty.** To encourage diversity within a training batch $\mathcal{X}$, we introduce a repetition penalty. We could use any similarity metric, but in our case, we specifically use the BLEU score for faster computation, as this calculation must be performed numerous times during the rollout process. We compute pairwise distances using BLEU score similarity, $d_{ij} = 1 - \text{BLEU}(x_i, x_j)$, and group questions where $d_{ij} < \tau_{\text{BLEU}}$ into clusters $\mathcal{C} = \{C_1, \ldots, C_K\}$. The penalty for a question $x_i$ in a cluster $C_k$ is proportional to its relative size:

$$r_{\text{rep}}(x_i) = \lambda \frac{|C_k|}{B}$$

where $B$ is the batch size and $\lambda$ is a scaling factor. In our experiments, we set $\lambda = 1$. The implementation details are shown in Appendix B.4.

**Format Check Penalty.** A critical first step in the reward pipeline is a structural format check to verify that each generated question is correctly enclosed within `<question>` and `</question>` tags. If the output does not adhere to this required structure, it is immediately assigned a final reward of 0, and no further reward signals are computed.

**Composite Reward and Policy Update.** For all questions that pass the format check, we calculate a composite reward. The final scalar reward $r_i$ for each valid question $x_i$ combines signals for uncertainty and repetition:

$$r_i = \max\big(0, r_{\text{uncertainty}}(x_i; \phi) - r_{\text{rep}}(x_i)\big)$$

With these rewards $\{r_1, \ldots, r_G\}$ for a batch of generated questions, we compute the advantage $\hat{A}_i$ for each question and update the Challenger's policy $Q_\theta$ by minimizing the GRPO loss $\mathcal{L}_{\text{GRPO}}(\theta)$.

## 2.3 Solver Dataset Construction

After updating the Challenger, we use it to generate a new, curated dataset to train the Solver. This process acts as a curriculum generator. We first sample a large pool of $N$ candidate questions from the Challenger's policy, $x_i \sim Q_\theta(\cdot \mid p_0)$. For each question, we obtain $m$ answers from the current Solver, determine the pseudo-label $\tilde{y}_i$ via majority vote, and calculate the empirical correctness $\hat{p}_i$. A question-answer pair $(x_i, \tilde{y}_i)$ is added to the training set $\mathcal{S}$ only if its correctness falls within an informative band, $|\hat{p}_i - \frac{1}{2}| \leq \delta$. This filtering step discards tasks that are either too easy or too hard.

While the primary goal of this filtering is to discard tasks that are too easy or too hard, it also serves as an implicit quality control mechanism. Since our pseudo-labels are derived from a majority vote, a very low empirical correctness $\hat{p}_i$ often indicates that the question itself is ambiguous, ill-posed, or that the resulting pseudo-label is unreliable. By filtering out these low-consistency items, our method simultaneously improves the quality and the uncertainty calibration of the training data.

## 2.4 Solver Training

The Solver, $S_\phi$, is then fine-tuned on the curated dataset of challenging problems $\mathcal{S}$. We also use GRPO for this stage, but with a simpler, verifiable reward signal. For a given question $x_i \in \mathcal{S}$ with its pseudo-label $\tilde{y}_i$, the Solver generates a batch of answers, each assigned a binary reward $r_j$:

$$r_j = \begin{cases} 1, & \text{if } x_j \text{ matches with the pseudo-label } \tilde{y}_i, \\ 0, & \text{otherwise.} \end{cases}$$

This verifiable reward is used to compute the advantage $\hat{A}_j$, and the Solver's policy $S_\phi$ is subsequently updated by minimizing the GRPO loss $\mathcal{L}_{\text{GRPO}}(\phi)$. This process enhances the Solver's ability to correctly answer the difficult questions generated by its co-evolving Challenger.

# 3 Experiments

## 3.1 Models and Training Details

We employ the Qwen3-4B-Base (Yang et al., 2025) and Qwen3-8B-Base models to assess the impact of scale within a single architectural family. Second, to ensure our approach is effective on a distinct lineage, we utilize the OctoThinker-3B and OctoThinker-8B models (Wang et al., 2025b).This choice is particularly relevant as Wang et al. (2025b) reported that applying RL training directly to Llama models yielded suboptimal results. As the OctoThinker series is continually trained from the Llama-3.1 models (Dubey et al., 2024), this comprehensive selection allows us to test our framework across different foundational architectures – Qwen vs. Llama. We assess our framework on a comprehensive suite of benchmarks, with the evaluation benchmarks presented in Appendix C.

Our entire framework is implemented based on the EasyR1 codebase (Zheng et al., 2025b). In each iteration of the *R-Zero* co-evolutionary loop, we follow a specific set of hyperparameters. The Challenger ($Q_\theta$) first generates a candidate pool of $N = 8,000$ questions. To construct the training dataset for the Solver, these questions are filtered based on consistency. For each candidate question, we sample $m = 10$ answers from the current Solver ($S_\phi$). A question is retained for the training set only if the number of answers matching the majority-vote pseudo-label is between 3 and 7, inclusive ($\delta = 0.25$). This numerical range is consistent with the methodology used in previous research (Zhang & Zuo, 2025; Li et al., 2025b; Bercovich et al., 2025). When training the Challenger, the uncertainty reward $r(x; \phi)$ is calculated by sampling $m = 10$ responses from the Solver. For the intra-batch repetition penalty, we set the clustering distance threshold to $\tau_{\text{BLEU}} = 0.5$.

Table 1: Comprehensive results on mathematical reasoning benchmarks. We compare each base model against a *R-Zero* (❄ challenger) baseline (where the Solver is trained on questions from an untrained Challenger) and our method, *R-Zero*. The peak performance is highlighted in **bold**.

| Model Name | Avg. | AMC | Minerva | MATH | GSM8K | Olympiad | AIME25 | AIME24 |
|---|---|---|---|---|---|---|---|---|
| *Qwen3-4B-Base* | | | | | | | | |
| Base Model (w/o training) | 42.57 | 45.70 | 38.24 | 68.20 | 87.79 | 41.04 | 10.30 | 6.7 |
| Absolute Zero | 46.42 | 52.45 | 41.96 | 76.20 | 89.34 | 42.56 | 10.20 | 12.20 |
| *R-Zero* (❄ challenger) | 45.01 | 45.00 | 45.22 | 72.80 | 87.87 | 41.19 | **10.20** | 12.80 |
| ***R-Zero* (our method)** | **49.93** | **57.27** | **52.94** | **79.60** | **92.12** | **44.59** | 9.60 | **13.40** |
| *Qwen3-8B-Base* | | | | | | | | |
| Base Model (w/o training) | 48.64 | 51.95 | 50.00 | 78.00 | 89.08 | 44.74 | 12.10 | 14.60 |
| Absolute Zero | 52.68 | 57.89 | 57.90 | 76.60 | 92.00 | 47.80 | 18.20 | 18.40 |
| *R-Zero* (❄ challenger) | 52.10 | 60.70 | 57.72 | 81.60 | 92.56 | 46.44 | 11.60 | 14.10 |
| ***R-Zero* (our method)** | **53.72** | **61.67** | **60.66** | **82.00** | **94.09** | **48.89** | 13.30 | 15.40 |
| *OctoThinker-3B* | | | | | | | | |
| Base Model (w/o training) | 26.64 | 17.19 | 24.26 | **55.00** | 73.69 | 16.15 | 0.21 | 0.00 |
| Absolute Zero | 27.23 | 22.50 | 22.70 | 53.20 | 75.80 | 13.20 | 0.50 | 2.70 |
| *R-Zero* (❄ challenger) | 27.51 | 20.19 | 24.63 | 54.60 | **74.98** | 15.70 | 0.10 | **2.40** |
| ***R-Zero* (our method)** | **29.32** | **27.03** | **27.57** | 54.20 | **74.98** | **18.22** | **3.23** | 0.00 |
| *OctoThinker-8B* | | | | | | | | |
| Base Model (w/o training) | 36.41 | 32.11 | 41.91 | 65.20 | 86.96 | 26.52 | **1.56** | 0.62 |
| Absolute Zero | 36.60 | 32.50 | 44.90 | 62.80 | 87.00 | 25.60 | 3.30 | 0.10 |
| *R-Zero* (❄ challenger) | 36.98 | 29.30 | 42.28 | 66.20 | **88.10** | **27.56** | 1.04 | **4.38** |
| ***R-Zero* (our method)** | **38.52** | **34.03** | **48.22** | **68.80** | 87.19 | **27.56** | 0.42 | 3.44 |

### 3.1.1 EVALUATION SETTING

The evaluation code is adopted from General-Reasoner (Ma et al., 2025). To ensure consistency, we reran the released evaluation code and reported the corresponding results. The reproduced results are well aligned with General-Reasoner and those in the Qwen-3 technical report (Yang et al., 2025).

The results presented in all experimental tables are obtained after 45 training steps, while in the figures we report evaluations performed at checkpoints every 15 steps during solver training. All results are reported based on the held-out test sets, ensuring a fair comparison across baselines and reproduced methods. Further implementation details and prompts can be found in Appendix B.

### 3.2 RESULTS IN MATHEMATICAL REASONING

The comprehensive results of our experiments are presented in Table 1. The findings confirm that our proposed framework, *R-Zero*, is a highly effective, model-agnostic method for enhancing the performance of language models on mathematical tasks across different architectures and scales.

Our iterative training process consistently and substantially improves upon the performance of the base models. This holds true for large models like Qwen3-8B-Base, where three iterations of *R-Zero* raise the average performance from a baseline of 49.18 to 54.69, a significant gain of **+5.51** points. Similarly, on the smaller OctoThinker-3B, our method improves the average score from 26.64 to 29.32 (**+2.68** points), demonstrating the broad applicability of our self-supervised training loop.

The critical role of the Challenger's RL-based training is validated by the immediate performance leap from the Base Challenger to the first iteration of *R-Zero*. On Qwen3-8B-Base, this first iteration provides a +1.52 point gain over the baseline, and the improvement is even more pronounced on Qwen3-4B-Base at +3.7 points. This confirms that the intelligent curriculum generated by the RL-trained Challenger is significantly more effective than that of a non-trained generator.

### 3.3 RESULTS IN GENERAL REASONING

Previous work has demonstrated that training language models on reasoning-intensive domains, such as mathematics, can lead to improvements in general-domain capabilities (Huan et al., 2025). A key question, however, is whether this generalization effect still holds when the training curriculum is not human-labeled, but entirely self-generated through *R-Zero*.

Table 2: Results on general-domain reasoning benchmarks. The table compares the Base Model, a *R-Zero* (❄ challenger) baseline, and our *R-Zero*. The peak performance is highlighted in **bold**.

| Model Name | Overall Avg. | SuperGPQA | MMLU-Pro | BBEH |
|---|---|---|---|---|
| *Qwen3-4B-Base* | | | | |
| Base Model (w/o training) | 26.34 | 20.88 | 50.58 | 7.57 |
| Absolute Zero | 29.33 | 27.10 | 52.60 | 8.30 |
| *R-Zero* (❄ challenger) | 28.52 | 24.77 | 54.20 | 6.59 |
| ***R-Zero* (our method)** | **31.15** | **27.55** | **55.47** | **10.42** |
| *Qwen3-8B-Base* | | | | |
| Base Model (w/o training) | 31.98 | 28.33 | 58.97 | 8.63 |
| Absolute Zero | 34.40 | 31.89 | 60.50 | 10.80 |
| *R-Zero* (❄ challenger) | 31.29 | 30.12 | 54.14 | 9.60 |
| ***R-Zero* (our method)** | **34.50** | **31.38** | **61.53** | **10.60** |
| *OctoThinker-3B* | | | | |
| Base Model (w/o training) | 7.47 | 10.09 | 10.87 | 1.46 |
| Absolute Zero | 16.03 | 17.70 | 24.30 | 6.10 |
| *R-Zero* (❄ challenger) | 10.04 | 11.19 | 14.53 | **4.40** |
| ***R-Zero* (our method)** | **11.12** | **12.44** | **16.71** | 4.20 |
| *OctoThinker-8B* | | | | |
| Base Model (w/o training) | 11.70 | 13.26 | 20.21 | 1.64 |
| Absolute Zero | 18.40 | 18.80 | 31.40 | 5.00 |
| *R-Zero* (❄ challenger) | 21.30 | 16.99 | **41.46** | 5.46 |
| ***R-Zero* (our method)** | **23.00** | **19.82** | 40.92 | **8.25** |

As shown in Table 2, this transfer of skills is evident across all tested models. For instance, three iterations of our math-focused training improve the average general-domain score of Qwen3-8B-Base by +5.13 points and OctoThinker-3B by +3.65 points. This generalization also extends to the key performance patterns observed in the mathematical results. This confirms that our method does not merely teach domain-specific knowledge, but enhances the model's underlying capabilities in a way that successfully generalizes across domains.

## 4 ANALYSIS

In this section, we conduct a series of in-depth analyses to better understand the behavior and effectiveness of our *R-Zero* framework. To ensure consistency, all analytical experiments presented here are conducted on the Qwen3-4B-Base model, unless explicitly stated otherwise. Some additional analyses are shown in Appendix D.

### 4.1 ABLATION STUDY

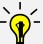 Removing Repetition Penalty and Task Filtering will harm the final performance.

To isolate the contribution of each key component within our *R-Zero* framework, we conduct a comprehensive ablation study on the `Qwen3-4B-Base` model. We specifically investigate the importance of two critical modules by disabling them one at a time and observing the impact on performance. The results are summarized in Table 3.

As shown in the table, removing any core components leads to a significant degradation in performance. Removing the **Repetition Penalty** harms performance, indicating that generating a diverse set of questions is crucial for effective Solver training.

Table 3: Ablation results on Qwen3-4B-Base. **w/o Filtering**: Disables the difficulty-based curriculum filtering. **w/o Rep. Penalty**: Removes the repetition penalty from the Challenger's reward.

| Method | Math AVG | General AVG |
|---|---|---|
| *R-Zero* | 49.07 | 31.15 |
| *Ablations* | | |
| ⊢ w/o Rep. Penalty | 45.76 | 28.73 |
| ⊢ w/o Filtering | 47.35 | 26.69 |

Finally, disabling the **Task Filtering** module results in a notable performance drop, particularly on the general-domain average, which falls by over 6 points. As discussed in Section 2.3, this filtering serves a dual purpose: it calibrates the curriculum's difficulty and acts as an implicit quality control mechanism by removing questions with low answer consistency. Without this filter, the Solver is trained on noisy data that likely includes ambiguous or ill-posed questions, which harms its ability to learn robustly.

## 4.2 ITERATION SCALING

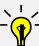 Model performance eventually converges, with the timing depending on model size.

Previous results demonstrate that *R-Zero* enhances the Solver's capabilities. This raises a critical question about the long-term stability of our self-improvement loop: *what are the limits of this process, and whether the eventual performance degradate?* In this section, we conduct an analysis to investigate these iteration scaling dynamics, aiming to diagnose the underlying cause of this instability.

As illustrated in Figure 3, our framework initially delivers on its promise, with models of all sizes showing significant performance improvements in the early stages of co-evolution. Unfortunately, this virtuous cycle does not continue indefinitely. After multiple iterations, we observe a consistent and concerning trend of performance degradation across all models. Intriguingly, we found a direct correlation between model scale and resilience to this collapse: the larger the model, the later the onset of performance degradation.

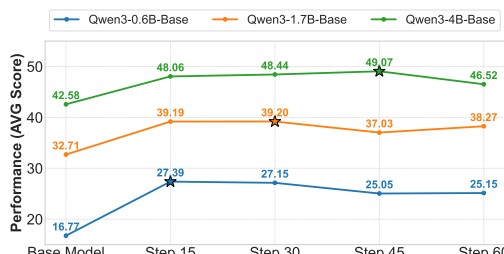

Figure 3: Math performance across different iteration times and model scales. The star markers indicate the peak performance for each model size.

For instance, the smallest 0.6B model reaches its peak performance as early as the first iteration (Step 15), after which its capabilities begin to decline. In contrast, the largest 4B model sustains its upward trajectory for three full iterations, only experiencing a sharp drop at Step 60. This pattern strongly suggests that while larger model capacity can delay the negative effects, it does not prevent them. This eventual collapse points to an inherent instability or limitation within our current self-improvement framework, highlighting a critical area for future investigation. We present some additional analysis results for this in Appendix E.

## 4.3 SYNERGY WITH SUPERVISED DATA

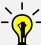 Using R-zero as a mid-training method boosts the effect of later training on human data.

To analyze the utility of our framework in scenarios where a labeled dataset is available, we measure the synergy between *R-Zero* and traditional supervised fine-tuning using labeled datasets [1]. The GRPO settings for this experiment were kept identical to our main experiments.

We first establish a supervised baseline by fine-tuning the base model directly on the labeled data. For this process, we also employ GRPO.

We then apply our *R-Zero* framework, where at the end of each co-evolutionary iteration, the resulting checkpoint is also fine-tuned on the same labeled dataset. The results show that our method provides significant additional gains. As highlighted in Figure 4, this represents a gain of **+2.35 points** over the human-label-only baseline for 4B model.

This finding confirms that *R-Zero* is not redundant with labeled data; instead, it acts as a powerful performance amplifier. The co-evolutionary process enables the model to better leverage the supervised information and achieve performance levels unattainable by standard fine-tuning alone.

---

[1] https://huggingface.co/datasets/hiyouga/math12k

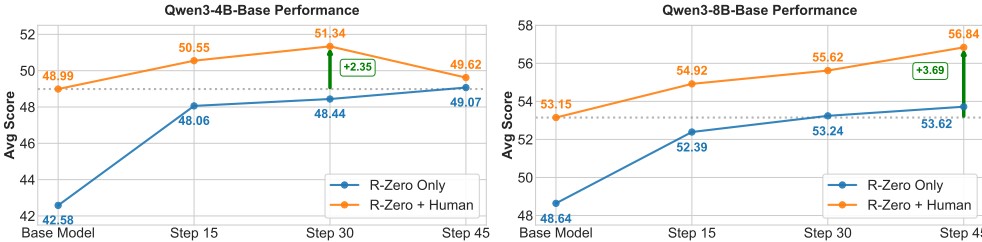

Figure 4: Performance of *R-Zero* when combined with supervised fine-tuning. The dashed line represents the baseline of fine-tuning the base model on labelled data alone, showing that our iterative method provides a better initialization.

We also studied the scenario where human-labeled datasets are mixed into the R-Zero generated dataset for training. We found that this concurrent training approach outperforms using either R-Zero dataset or humanlabel dataset alone, but it does not surpass the sequential strategy of first applying R-Zero and then performing SFT. We hypothesize that mixing the labeled data during R-Zero training may dilute the high-quality signal while partially mitigating noise. Performing R-Zero first allows the model to acquire reasoning ability before leveraging the high-quality human-labeled data, which appears to be the most effective strategy. The precise underlying reasons require further investigation.

| Dataset | AMC | Minerva | MATH | GSM8K | Olympiad | AIME25 | AIME24 | SuperGPQA | MMLU-Pro | BBEH |
|---|---|---|---|---|---|---|---|---|---|---|
| Human | 57.97 | 55.15 | 80.8 | 92.04 | 48 | 9.58 | 10.31 | 29.49 | 57.03 | 9.71 |
| *R-Zero* | 57.27 | 52.94 | 79.6 | 92.12 | 44.59 | 9.6 | 13.4 | 27.55 | 55.47 | 10.42 |
| *R-Zero*+Human | 57.9 | 53.2 | 81.2 | 92.2 | 47.7 | 10.32 | 14.67 | 29.4 | 58.2 | 11.8 |

Table 4: Performance comparison across benchmarks when training with human-labeled data, R-zero self-generated data, and a mixed training strategy.

## 4.4 EVOLUTION OF QUESTION DIFFICULTY AND DATA ACCURACY

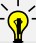 Question difficulty increases progressively with decreasing pseudo-label accuracy.

Table 5: Performance and data accuracy analysis. The highlighted column represents the *true accuracy* of the self-generated pseudo-labels for each question set.

| | Performance of Evaluated Model (vs. Ground Truth) | | | | |
|---|---|---|---|---|---|
| | Base Model | Solver (step 15) | Solver (step 30) | Solver (step 45) | Pseudo-Label Acc. |
| $\mathcal{D}_{\text{Step 15}}$ | 48.0 | 59.0 | 57.0 | 61.0 | 79.0% |
| $\mathcal{D}_{\text{Step 30}}$ | 52.5 | 53.0 | 51.5 | 53.5 | 69.0% |
| $\mathcal{D}_{\text{Step 45}}$ | 44.0 | 47.0 | 45.0 | 50.5 | 63.0% |

To understand the co-evolutionary dynamic, we analyzed how the Challenger's generated questions and their corresponding pseudo-labels change across iterations. We sampled 200 questions from the Challenger's policy after each of the first three training iterations, creating three distinct test sets: $\mathcal{D}_{\text{step 15}}$, $\mathcal{D}_{\text{step 30}}$, and $\mathcal{D}_{\text{step 45}}$. For this analysis, we assumed the external oracle model, GPT-4o, to be a perfect annotator, providing the ground truth answers for all generated questions.

The evaluation was conducted as follows: the performance of our internal models was measured against these GPT-4o ground truth answers. The score reported for GPT-4o itself, however, reflects the **true accuracy of our self-generated pseudo-labels** by comparing the pseudo label against the ground truth from the oracle (GPT-4o). The results on the filtered dataset are summarized in Table 5.

This analysis reveals a multi-faceted dynamic. The first finding is that the questions generated by the Challenger become **progressively more difficult**. This is directly evidenced by evaluating a fixed

model against the evolving question sets. For instance, the performance of the static Solver (Step 15), when measured against the consistent GPT-4o ground truth, drops from 59.0% on the questions for the Step 15 training to 47.0% on the questions for the Step 45. This confirms that the Challenger is successfully increasing the intrinsic difficulty of its curriculum.

The second finding, revealed by the highlighted column, pertains to the **true accuracy of the self-generated dataset**. Unfortunately, while the accuracy of the pseudo-labels is initially high at 79.0%, it systematically drops to 63.0% by the third iteration. This trend indicates that as the system generates more difficult problems, the Solver's majority vote becomes a less reliable source for ground truth. This decline in data quality is a critical trade-off and a potential bottleneck for the framework's ultimate performance.

## 5 RELATED WORK

### 5.1 LABEL-FREE REINFORCEMENT LEARNING

A significant trend in recent research is Label-Free Reinforcement Learning, which aims to improve LLM reasoning without human-annotated data. Many such methods use the model's own outputs as a reward signal. This includes leveraging sequence-level confidence (Li et al., 2025a; Prabhudesai et al., 2025), the consistency of answers derived from varied reasoning paths (Zhang et al., 2025a; Zuo et al., 2025; Zhang et al., 2025b; Zhou et al., 2025b; Prasad et al., 2024), minimizing the output entropy (Agarwal et al., 2025; Cheng et al., 2025), or even random (Shao et al., 2025) or negative reward (Zhu et al., 2025). These signals are often used within self-training loops where models fine-tune on their own most plausible solutions (Shafayat et al., 2025; Zhao et al., 2025b). While these methods all rely on a pre-existing set of unlabeled problems, *R-Zero* removes the need for any seed dataset.

### 5.2 SELF-PLAY IN LARGE LANGUAGE MODELS

The paradigm of self-play, where models take on dual roles to create a self-improvement loop (Chen et al., 2024; Zhang et al., 2024), has recently been adapted to improve language models without human data. This approach has been particularly fruitful in verifiable domains like code generation, where a "Coder" agent's program is verified by a "Tester" agent's unit tests (Lin et al., 2025; Wang et al., 2025a; Pourcel et al., 2025; Jiang et al., 2025). More advanced frameworks push autonomy further by learning to generate the problems themselves, creating an adaptive curriculum from a small seed of examples or from scratch (Zhao et al., 2025a; Li et al., 2025c; Zhou et al., 2025a; Fang et al., 2025). Our work distinguishes itself by extending this paradigm to general reasoning domains that lack such verifiable environments, like coding tasks.

### 5.3 REINFORCEMENT LEARNING WITH VERIFIABLE REWARDS (RLVR)

Reinforcement Learning with Verifiable Rewards has been widely adopted as a versatile paradigm for enhancing LLMs across a multitude of tasks (Li et al., 2025d; DeepSeek-AI et al., 2025; Shao et al., 2024). Its effectiveness is demonstrated in diverse applications such as relation extraction (Dai et al., 2025), interactive GUI navigation (Shi et al., 2025b) and search-engine utilization (Jin et al., 2025). While early implementations relied on rule-based verifiers, recent work has begun to explore more sophisticated, model-based verifiers (Ma et al., 2025; Li et al., 2025b; 2024).

## 6 LIMITATION

While R-Zero demonstrates strong improvements in reasoning performance across multiple domains, several limitations remain. First, the core mechanism of R-Zero relies on domains where correctness can be objectively verified. The Challenger–Solver co-evolution process depends on clear, deterministic evaluation signals to produce reliable training feedback. Consequently, applying R-Zero to open-ended or subjective tasks, such as creative writing, dialogue, or preference-driven generation, remains difficult, as these tasks lack unambiguous correctness criteria. In addition, R-Zero currently employs specific labeling and verification strategies that may not generalize to all task

types. Developing more robust and broadly applicable labeling mechanisms would further expand the range of domains where R-Zero can be effectively applied.

## 7    CONCLUSION AND FUTURE WORK

In this paper, we introduced *R-Zero*, a fully autonomous self-evolving framework that overcomes data dependency by having a Challenger and Solver co-evolve to create a self-generating curriculum. Our experiments demonstrate that *R-Zero* significantly improves LLM's reasoning capability in multiple domains. Future work could further focus on improving efficiency, exploring more robust labeling techniques, and expanding *R-Zero* to new domains. Extending this self-evolutionary paradigm to open-ended generative tasks, such as creative writing or dialogue, where evaluation is subjective, remains a significant hurdle for future research.

## REPRODUCIBILITY STATEMENT

To ensure the reproducibility of our research, we provide detailed information regarding our data and experimental setup. All datasets used in this work are publicly available; we provide details on data sources, any postprocess steps in Appendix B. A comprehensive list of all hyperparameters for each experiment can be found in a table in Appendix B and Sec. 3.

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

APPENDIX

# A    THE USE OF LARGE LANGUAGE MODELS (LLMS)

We acknowledge the use of large language models (LLMs) as assistive tools in this research, with usage limited to refining grammar and improving language clarity in the manuscript, writing utility scripts for data preprocessing and postprocessing, and debugging; all outputs from these models were meticulously reviewed, revised, and verified by the authors, who retain full responsibility for all content presented in this paper.

# B    EXPERIMENT DETAILS

## B.1    TRAINING HYPERPARAMETER

This section summarizes the most critical algorithmic hyperparameters for the Solver and Challenger training stages. All experiments were conducted using BFloat16 (BF16) mixed precision and FlashAttention 2.

### B.1.1    SOLVER TRAINING

- **Global Batch Size**: 128
- **Learning Rate**: $1 \times 10^{-6}$
- **Weight Decay**: $1 \times 10^{-2}$
- **KL Penalty Coefficient** ($\lambda_{KL}$): $1 \times 10^{-2}$
- **Max Steps**: 15
- **Number of Rollouts**: 5
- **Rollout Temperature**: 1.0
- **Rollout Top-p**: 0.99

### B.1.2    CHALLENGER TRAINING

- **Global Batch Size**: 128
- **Learning Rate**: $1 \times 10^{-6}$
- **Weight Decay**: $1 \times 10^{-2}$
- **KL Penalty Coefficient** ($\lambda_{KL}$): $1 \times 10^{-2}$
- **Max Steps**: 5
- **Number of Rollouts**: 4
- **Rollout Temperature**: 1.0
- **Rollout Top-p**: 0.99

## B.2    PROMPT TEMPLATES

This section presents the exact prompt templates used for the solver and challenger models.

> **Solver Prompt Template**
>
> **System Message:**
> Please reason step by step, and put your final answer within \boxed{}.
> **User Message:**
> {*problem_statement*}
>
> *Note:* {`problem_statement`} *is a placeholder for the actual math problem.*

---

**Challenger Prompt Template**

**System Message:**
You are an expert competition-math problem setter. FIRST, in your private scratch-pad, think step-by-step to design a brand-new, non-trivial problem. The problem could come from any field of mathematics, including but not limited to algebra, geometry, number theory, combinatorics, prealgebra, probability, statistics, and calculus. Aim for a difficulty such that fewer than 30% of advanced high-school students could solve it. Avoid re-using textbook clichés or famous contest problems.
THEN, without revealing any of your private thoughts, output **exactly** the following two blocks:

```
<question>
{The full problem statement on one or more lines}
</question>

\boxed{final_answer}
```

Do NOT output anything else—no explanations, no extra markup.
**User Message:**
Generate one new, challenging reasoning question now. Remember to format the output exactly as instructed.

---

### B.3 GPT-4O JUDGE PROMPT

To programmatically evaluate the correctness of answers on mathematical benchmarks where the final answer can be complex (e.g., simplified expressions), we use GPT-4o as a judge. The exact prompt and configuration used for this evaluation are detailed below.

---

**Configuration for GPT-4o as Judge**

- **Model**: gpt-4o
- **Temperature**: 0.1

**System Message:**

> You are a math answer checker.

**User Message Template:**

> Hi, there is an answer: {answer},
> and the ground truth answer is: {response},
> please check whether the answer is correct or not, and return the **only**
> Yes or No.

*Note: {answer} is a placeholder for the model-generated solution, and {response} is the ground-truth answer from the benchmark.*

---

### B.4 REPETITION PENALTY IMPLEMENTATION

To encourage the Challenger to generate a diverse set of questions within each batch, we apply a repetition penalty, $r_{\text{rep}}$. This penalty is designed to disincentivize the model from producing semantically similar questions in the same batch. The implementation is a multi-step process based on clustering questions by their BLEU score similarity.

**1. Pairwise Distance Calculation via BLEU Score** First, we compute a pairwise distance matrix for all questions in a batch. The distance $d_{ij}$ between any two questions, $x_i$ and $x_j$, is defined as one minus their BLEU score:

$$d_{ij} = 1 - \text{BLEU}(x_i, x_j)$$

For this calculation, we specifically use the `sentence_bleu` function from the NLTK library (`nltk.translate.bleu_score`). To ensure numerical stability, especially for shorter questions with limited n-gram overlap, we employ its first smoothing function, `SmoothingFunction().method1`. The questions are tokenized for the BLEU calculation by splitting on whitespace; no further text normalization, such as lowercasing or punctuation removal, is performed.

**2. Agglomerative Clustering** With the pairwise distance matrix computed, we then group similar questions using agglomerative hierarchical clustering. This step is performed using the `Clustering` implementation from the `scikit-learn` library. The clustering algorithm is configured with the following key parameters:

- **Metric**: Set to `'precomputed'`, indicating that we provide our custom BLEU-based distance matrix instead of having the algorithm compute distances.
- **Linkage**: Set to `'average'`. This method defines the distance between two clusters as the average of the distances between all pairs of questions across the two clusters.

**3. Final Penalty Calculation** Once each question in the batch is assigned to a cluster, the repetition penalty $r_{\text{rep}}(x_i)$ for a given question $x_i$ is determined by the relative size of the cluster $C_k$ to which it belongs. The penalty is calculated as:

$$r_{\text{rep}}(x_i) = \frac{|C_k|}{B}$$

Here, $|C_k|$ represents the number of questions in cluster $C_k$, and $B$ is the total number of questions in the batch (i.e., the batch size).

## C  Evaluation Benchmark

We assess our framework on a comprehensive suite of benchmarks. Although the question-generator prompt for our method is primarily focused on mathematical problem-solving, a key objective of our evaluation is to explore whether the resulting improvements in reasoning ability can generalize to other domains. Therefore, our evaluation is divided into two main categories.

**Mathematical Reasoning.** We use seven challenging benchmarks: AMC, Minerva (Lewkowycz et al., 2022), MATH-500 (Hendrycks et al., 2021b), GSM8K (Cobbe et al., 2021), Olympiad-Bench (He et al., 2024), AIME-2024, and AIME-2025. For these tasks, where answers can be complex, we employ GPT-4o as a programmatic judge to semantically verify the correctness of the final answer against the ground truth (Zhao et al., 2025c). For the difficult AMC and AIME benchmarks, we report the **mean@32** metric. For all other math benchmarks, we report accuracy based on greedy decoding.

**General Domain Reasoning.** To test for the generalization of reasoning ability, we evaluate the following challenging benchmarks:

- **MMLU-Pro** (Wang et al., 2024): An enhanced version of the MMLU (Hendrycks et al., 2021a) benchmark, featuring a more challenging suite of multi-task questions designed to provide a stricter evaluation of language model capabilities.
- **SuperGPQA** (Du et al., 2025): A large-scale benchmark focused on graduate-level reasoning. It comprises questions across 285 distinct disciplines that have been verified as unsearchable on the web, thereby isolating true reasoning ability from simple knowledge recall.
- **BBEH** (shoaa kazemi et al., 2025): This benchmark builds upon the foundation of BIG-Bench Hard (Suzgun et al., 2023) by incorporating a new selection of tasks specifically engineered to be more difficult, thus providing a more accurate measure of complex reasoning skills.

For this category, we follow the experimental setup, prompts, and evaluation codes from (Ma et al., 2025), reporting Exact Match (EM) accuracy obtained via greedy decoding.

# D  PARAMETER SHARING BETWEEN CHALLENGER AND SOLVER

> 💡 Separating the Challenger and Solver into two models will be better.

Table 6: Comparison of math performance and pseudo-label accuracy between the standard R-Zero (two-model) and Single-R-Zero (unified model, shared parameters) frameworks across iterations.

| Iteration | R-Zero (ours) | | Single-R-Zero | |
|---|---|---|---|---|
| | Performance | Pseudo-label Acc (%) | Performance | Pseudo-label Acc (%) |
| Step 15 | 48.06 | 71.0 | **47.31** | 63.4 |
| Step 30 | 48.44 | 56.2 | 46.95 | 46.6 |
| Step 45 | **49.12** | 48.8 | 45.57 | 32.6 |
| Step 60 | 46.52 | 42.2 | 43.89 | 33.8 |

To investigate whether the separation of the Challenger and Solver into two independent models is a necessary component for the success of *R-Zero*, we conduct an ablation study using a unified model with shared parameters. In this configuration (Single-R-Zero), a single model is tasked with performing both roles, i.e., generating a challenging curriculum and subsequently learning from it.

The results, presented in Table 6, clearly indicate that separating the Challenger and Solver into two independent models is crucial for both performance and stability. We observe two key findings. First, our standard two-model R-Zero framework not only achieves a higher peak performance (49.12) but also sustains improvement for more iterations, with its collapse occurring after the third iteration. In contrast, the unified Single-R-Zero model's performance peaks after the very first iteration and degrades immediately thereafter. Second, the Single-R-Zero model, where the agent must generate and solve its own problems, produces pseudo-labels of significantly lower accuracy at every stage. For example, in the first iteration, its pseudo-label accuracy is already substantially lower than the R-Zero's (63.4% vs. 71.0%). We hypothesize that this is because having the problem-setter and solver originate from the same model leads to a form of overconfidence that comes from internal bias.

# E  BEYOND LABEL NOISE: UNPACKING THE ROOTS OF INSTABILITY

The most immediate hypothesis for this performance collapse is the degradation of pseudo-label quality, a potential failure mode of the self-correction mechanism we discussed in Section 4.4. As the Challenger generates increasingly difficult problems, it is plausible that the Solver's majority vote becomes a less reliable source for ground truth, resulting in a noisy training signal that could ultimately harm performance. To empirically test the extent to which this is the primary cause, we sampled 500 questions from a later training iteration to conduct a more granular investigation into the relationship between pseudo-label fidelity and the observed performance drop.

Table 7: Accuracy of self-generated pseudo-labels (%), labeled by Gemini. Shaded and bolded values indicate the best checkpoint for each model size.

| Step | Model Size | | |
|---|---|---|---|
| | 0.6B | 1.7B | 4B |
| Step 15 | **70.6** | 69.4 | 71.0 |
| Step 30 | 53.4 | **55.2** | 56.2 |
| Step 45 | 50.8 | 52.2 | **48.8** |
| Step 60 | 44.0 | 45.2 | 42.2 |

Although the degradation of pseudo-label accuracy is a consistent trend across iterations, our analysis suggests this is not the primary, nor even the sole, driver of the eventual performance collapse. Table 7 presents the pseudo-label data quality for each model at the onset of its performance collapse. Intriguingly, there appears to be no universal accuracy threshold that triggers this degradation. For instance, the 0.6B model begins to decline when data accuracy is still as high as 70.6% (Step 15), whereas the 4B model tolerates an accuracy as low as 48.8% (Step 45) before its performance drops.

This suggests that the absolute percentage of label noise is not the sole determinant of instability. Another potential, and perhaps more fundamental, reason is a form of model collapse that can be introduced when training exclusively on self-synthesized data (Tan et al., 2024; Shumailov et al., 2024; Dohmatob et al., 2024b; Zhou et al., 2025c; Seddik et al., 2024; Dohmatob et al., 2024a;

Briesch et al., 2023; Zheng et al., 2025a). A model can enter a degenerative feedback loop, suffering from a loss of diversity or an amplification of its own biases, which presents a significant challenge.

## F    THEORETICAL ANALYSIS

In this section, we provide a theoretical motivation for our uncertainty reward function, $r_{\text{uncertainty}} \propto 1 - 2|\hat{p}(x; S_\phi) - \frac{1}{2}|$, which is maximized when the Solver's success probability, $\hat{p}$, is 50%. Our analysis is grounded in recent work that formally establishes that the most efficient training occurs when a learner is exposed to tasks at the frontier of its capabilities (Shi et al., 2025a; Bae et al., 2025).

The core insight from these studies is that the learning potential of the current Solver, with policy $S_\phi$, can be quantified by the KL divergence to an optimal policy $S^*$. This divergence, $\mathbb{D}_{KL}(S_\phi||S^*)$, is lower-bounded by the variance of the Solver's reward. For the binary reward signal used in our framework, the success probability is $\hat{p}$. This leads to the specific lower bound:

$$\mathbb{D}_{KL}(S_\phi||S^*) \geq \frac{\hat{p}(1 - \hat{p})}{2\beta^2}$$

where $\beta$ is the temperature parameter that controls entropy regularization. The right-hand side of the inequality, which is proportional to the reward variance, is maximized precisely when $\hat{p} = 0.5$. Therefore, by designing the Challenger's reward to incentivize questions that push the current Solver towards this point of maximum uncertainty, our framework is theoretically motivated to generate a maximally efficient curriculum in each iteration of the co-evolutionary process.

# G  ALGORITHM OF *R-Zero*

Here, we provide the pseudocode for our method.

---

**Algorithm 1:** *R-Zero*: Self-Evolving Challenger-Solver Framework

---

**Input:** Initial models $Q_\theta, S_\phi$; Group size $G$; Solver samples $m$; Dataset size $N$; Filtering threshold $\delta$; Repetition threshold $\tau_{\text{BLEU}}$. **Output:** Evolved models $Q_\theta$ and $S_\phi$.

**for** *each self-play iteration* **do**

    // --- Phase 1: Challenger Training (Sec 2.2) ---

    **for** *each Challenger training step* **do**

        Sample question group $\{x_i\}_{i=1}^G \sim Q_\theta(\cdot)$;

        **for** *each question $x_i$* **do**

            **if** *FormatCheck($x_i$) is invalid* (e.g., missing tags) **then**

                $r_i \leftarrow 0$;

            **end**

            **else**

                Sample $m$ answers $\{y_j\}_{j=1}^m \sim S_\phi(\cdot \mid x_i)$;

                Get pseudo-label $\tilde{y}_i \leftarrow \text{MajorityVote}(\{y_j\}_{j=1}^m)$;

                Compute correctness $\hat{p}_i \leftarrow \frac{1}{m}\sum_{j=1}^m \mathbb{1}\{y_j = \tilde{y}_i\}$;

                Compute uncertainty reward $r_{\text{uncertainty}} \leftarrow 1 - 2|\hat{p}_i - \frac{1}{2}|$;

                Compute repetition penalty $r_{\text{rep}}(x_i)$ via BLEU clustering (Sec 2.2);

                Final reward: $r_i \leftarrow \max\big(0, r_{\text{uncertainty}} - r_{\text{rep}}(x_i)\big)$;

            **end**

        **end**

        Update $Q_\theta$ via GRPO using rewards $\{r_i\}_{i=1}^G$;

    **end**

    // --- Phase 2: Solver Dataset Construction (Sec 2.3) ---

    Initialize curated dataset $\mathcal{S} \leftarrow \emptyset$;

    Sample $N$ candidate questions $\{x_k\}_{k=1}^N \sim Q_\theta(\cdot)$;

    **for** *each candidate $x_k$* **do**

        Sample $m$ answers $\{y_j\}_{j=1}^m \sim S_\phi(\cdot \mid x_k)$;

        Get pseudo-label $\tilde{y}_k \leftarrow \text{MajorityVote}(\{y_j\})$;

        Compute correctness $\hat{p}_k \leftarrow \frac{1}{m}\sum_{j=1}^m \mathbb{1}\{y_j = \tilde{y}_k\}$;

        **if** $|\hat{p}_k - \frac{1}{2}| \le \delta$ **then**

            Add $(x_k, \tilde{y}_k)$ to $\mathcal{S}$;

        **end**

    **end**

    // --- Phase 3: Solver Training (Sec 2.4) ---

    **for** *each minibatch $(x, \tilde{y}) \in \mathcal{S}$* **do**

        Sample $G$ answers $\{y_j\}_{j=1}^G \sim S_\phi(\cdot \mid x)$;

        Compute binary rewards $r_j \leftarrow \mathbb{1}(y_j = \tilde{y})$;

        Update $S_\phi$ via GRPO using rewards $\{r_j\}_{j=1}^G$;

    **end**

**end**

---

# H  GENERALIZING BEYOND THE MATH DOMAIN

To further evaluate the generality of our proposed approach, we conduct an additional experiment in which the model is no longer instructed to generate exclusively mathematical problems during data construction. Instead, we remove the math-specific constraint from the prompting stage, allowing the model to produce a broader mixture of questions, including commonsense, logical reasoning, and other non-mathematical categories. This setup enables us to examine whether the improvements observed in earlier experiments are merely a consequence of alignment with math-heavy benchmarks or whether our method can maintain (or even enhance) performance under a more domain-general training regime.

Table 8: Results of different models and prompting strategies on mathematics and reasoning benchmarks.

| | AMC | Minerva | MATH | GSM8K | Olympiad | AIME25 | AIME24 | SuperGPQA | MMLU-Pro | BBEH |
|---|---|---|---|---|---|---|---|---|---|---|
| Qwen3-4B | 45.7 | 38.24 | 68.2 | 87.79 | 41.04 | 10.3 | 6.7 | 20.88 | 37.38 | 7.57 |
| R-Zero | 57.27 | 52.94 | 79.6 | 92.12 | 44.59 | 9.6 | 13.4 | 27.55 | 55.47 | 10.42 |
| R-Zero (no math) | 56.27 | 52.56 | 79.2 | 92.56 | 43.56 | 9.9 | 13.2 | 28.14 | 54.75 | 10.31 |
| Qwen3-8B | 51.95 | 50 | 78 | 89.08 | 44.74 | 12.1 | 14.6 | 28.33 | 51.8 | 8.63 |
| R-Zero | 61.67 | 60.66 | 82 | 94.09 | 48.89 | 13.3 | 15.4 | 31.38 | 61.53 | 10.60 |
| R-Zero (no math) | 65.53 | 61.19 | 81.6 | 93.89 | 49.24 | 13.2 | 15.2 | 32.29 | 61.83 | 10.88 |

The motivation behind introducing this experiment stems from concerns that many mathematical benchmarks contain well-structured and heavily studied problem formats. If the training data is restricted to similar mathematical tasks, it may artificially inflate performance gains while providing limited evidence regarding the method's real-world applicability. By diversifying the data generation process, we aim to assess whether the model can benefit from richer and more heterogeneous supervision signals.

The experimental results, summarized in Table 8, demonstrate that when the base model has sufficient capacity (e.g., the 8B variant), training on more general data leads to further improvements in both math and non-math benchmarks. Notably, the "R-Zero (remove math prompt)" setting exhibits the highest performance on the general benchmarks for the 8B model, suggesting that our approach effectively transfers to broader reasoning tasks beyond pure mathematics.

These findings provide additional evidence that the proposed method is not limited to mathematical domains; rather, it scales effectively with model capacity and supports improved reasoning performance under more diverse training conditions.

# I PRELIMINARIES

Our work builds upon recent advancements in reinforcement learning for fine-tuning large language models. We briefly review two key methodologies that are relevant to our framework.

## I.1 GROUP RELATIVE POLICY OPTIMIZATION

Group Relative Policy Optimization (GRPO) (Shao et al., 2024) is a reinforcement learning algorithm for fine-tuning policy LLMs $\pi_\theta$ without a separately learned value function. Its key idea is to normalize rewards within a group of responses sampled from the same prompt, thereby stabilizing optimization.

**Setup.** Given a query $\mathbf{q}$, the old policy $\pi_{\theta_{\text{old}}}$ generates $G$ candidate responses $\{\mathbf{o}_1, \ldots, \mathbf{o}_G\}$. Each response $\mathbf{o}_i$ is assigned a scalar reward $R(\mathbf{q}, \mathbf{o}_i)$. Group-wise z-score normalization is then applied to obtain an advantage shared across the tokens of the response:

$$\hat{A}_{i,t} = \frac{R(\mathbf{q}, \mathbf{o}_i) - \text{mean}\big(\{R(\mathbf{q}, \mathbf{o}_1), \ldots, R(\mathbf{q}, \mathbf{o}_G)\}\big)}{\text{std}\big(\{R(\mathbf{q}, \mathbf{o}_1), \ldots, R(\mathbf{q}, \mathbf{o}_G)\}\big) + \varepsilon_{\text{norm}}},$$

where $\varepsilon_{\text{norm}}$ is a small constant for numerical stability. The same normalized advantage $\hat{A}_{i,t}$ is applied to all tokens of response $\mathbf{o}_i$.

**Policy Update.** Let $r_\theta(o_{i,t}) = \frac{\pi_\theta(o_{i,t}|\mathbf{q}, \mathbf{o}_{i,<t})}{\pi_{\theta_{\text{old}}}(o_{i,t}|\mathbf{q}, \mathbf{o}_{i,<t})}$. The policy is updated with a clipped surrogate objective, similar to PPO, combined with a KL-divergence penalty to constrain policy drift:

$$\mathcal{L}_{\text{GRPO}}(\theta) = -\frac{1}{G}\sum_{i=1}^{G}\frac{1}{|\mathbf{o}_i|}\sum_{t=1}^{|\mathbf{o}_i|}\min\Big(r_\theta(o_{i,t})\,\hat{A}_{i,t},\,\text{clip}\big(r_\theta(o_{i,t}), 1-\epsilon, 1+\epsilon\big)\,\hat{A}_{i,t}\Big) + \beta\,\text{KL}\big(\pi_\theta(\mathbf{q})\,\|\,\pi_{\theta_{\text{old}}}(\mathbf{q})\big)$$

Maximizing the negative of this loss encourages the policy to increase the probability of tokens contributing to responses with positive relative advantages, while clipping prevents overly aggressive updates. The KL penalty, weighted by $\beta$, further stabilizes training by preventing the new policy from drifting too far from the old one.

## I.2 REINFORCEMENT LEARNING WITH VERIFIABLE REWARDS

Reinforcement Learning with Verifiable Rewards (RLVR) (Lambert et al., 2024) is a paradigm for fine-tuning models in domains where response quality can be deterministically verified. RLVR relies on a rule-based verifier $v : \mathcal{X} \rightarrow \{0, 1\}$ that assigns a binary reward to each response $x_i$:

$$r_i = v(x_i) = \begin{cases} 1, & \text{if } x_i \text{ satisfies a task-specific correctness check,} \\ 0, & \text{otherwise.} \end{cases}$$

This reward structure is especially effective for tasks like math, code generation with clear correctness criteria, and serves as the foundation for the reward mechanism in our Solver training.

