# OpenReview forum: "R-Zero: Self-Evolving Reasoning LLM from Zero Data"
_ICLR.cc/2026/Conference — ICLR 2026 Poster_

### Official Review · Reviewer_XCxR · 2025-10-21

**Soundness:** 2
**Presentation:** 3
**Contribution:** 2
**Rating:** 4
**Confidence:** 4

**Summary:**

This paper presents R-Zero, a framework for training reasoning LLMs through co-evolution of two models (Challenger and Solver) without requiring any human-curated data or labels. The approach uses reinforcement learning to create a self-improving curriculum where the Challenger generates increasingly difficult questions and the Solver learns to solve them. While the motivation is strong, there are numerous weaknesses that should be resolved before publication. While there is solid base model coverage (4 in total), the train domain is only 1 (math) and the synergy and use-case when labeled data is available is limited. Also, there is no existing approach baseline other than the base model performance, therefore, the authors should provide a strong rebuttal.

**Strengths:**

1. Paper is well motivated

2. Presentation is clear

3. Mid-training is an important direction for practitioners, therefore, this part is appreciated

> Our further analysis finds that R-Zero can act as a mid-training method, as models first improved by our method achieve higher performance after fine-tuned on labeled data.

4. Analysis section is appreciated

**Weaknesses:**

1. It is unclear if sufficient comparisons have been made with existing approaches, especially as there seems to be no baseline other than the base model. There should be significant label-free works that could serve as reasonable baselines. One that immediately comes to mind is Self-Consistency Preference Optimization.

2. a pseudocode to clearly follow this method step by step would be appreciated especially since it is a GAN-style back-and-forth going on. I believe that an intuitive view of how this algorithm works will be helpful. If there is no space it should at least be in the appendix. However, i think there is enough space if sec. 2 is removed. It is unclear why GRPO and RLVR needs to be introduced at this lvl of detail.

3. It is concerning that the training data is only on math domain. While i understand that various benchmarks have been used on the testing side, on the training side there should be more training domains to see generality of this approach. This is especially important since a lot of math benchmarks that you are seeing gains here already has labeled data, therefore, the experimental setup is less convincing from a practical view.

> We focus on generating questions specifically within the domain of mathematics, as it provides a convenient and self-contained setting for our framework; the objective nature of mathematical answers allows for the straightforward generation of pseudo-labels via majority voting, without the need for external verification environments like code executors.

4. There seems to be a lot of newly introduced hyperparameters. This does seem like a weakness for practical applications. Are these set heuristically? Arbitrarily? Are there any sensitivity tests for this?

> The Challenger (Qθ) first generates a candidate pool of N = 8, 000 questions. To construct the training dataset for the Solver, these questions are filtered based on consistency. For each candidate question, we sample m = 10 answers from the current Solver (Sϕ). A question is retained for the training set only if the number of answers matching the majority-vote pseudo-label is between 3 and 7, inclusive (δ = 0.25). This numerical range is consistent with the methodology used in previous research (Zhang & Zuo, 2025; Li et al., 2025b; Bercovich et al., 2025). When training the Challenger, the uncertainty reward r(x; ϕ) is calculated by sampling m = 10 responses from the Solver. For the intra-batch repetition penalty, we set the clustering distance threshold to τBLEU = 0.5

5. This method seems to make most sense for benchmarks that do not have a corresponding train set. In a practical perspective, there is no need to not use the train set of e.g. GSM8K. Either there should be meaningful empirical evidence on benchmarks with no training data, i.e., no labels. Or if you want to show that this approach is orthogonal, you should include regular SFT and GRPO as a baseline for benchmarks that have labeled data. Another approach would be to do SFT/GRPO on the existing train data + use your proposed approach to show that it adds gains. Without this, practical impact is not evidently clear. While i see that Sec. 5.3 exists, this kind of practical use when labeled data is available should have been available in the main experiments, as this is very important for practical use.

6. Contents of Sec. 5.3 should be expanded even at the cost of removing e.g. sec. 2.

7. There should be a clear analysis on the cost structure of this approach. I am guessing that the compute overhead over regular SFT is significantly larger as many samples needs to sampled and there are additional compute overhead for all these metrics, clustering, etc. the method itself is quite complex.

8. The top right part of Fig. 1 is not intuitive. My suggestion is using tick and X mark or something similar instead of color shades. Some readers might read the paper in black-and-white

9. It would be nice to have labels for the light blue and dark blue for the grouped bar chart in Fig. 1

10. It would be nice if there was a clear limitation section for practitioners.

**Questions:**

1. Are these methods in the baseline, if applicable? one of my main concern is the lack of baselines, hopefully the authors can clearly rebute this.

> self-challenging approaches train LLMs on tasks generated by the models themselves (Zhou
et al., 2025a; Wang et al., 2025a; Zhao et al., 2025a).

---

> ### Author Response · Authors · 2025-11-19
> **Response to Reviewer XCxR (Part 1)**
>
> Thank you very much for your thoughtful feedback, especially the many detailed comments on the presentation. We found them extremely helpful in improving the clarity of the paper. We have incorporated your suggestions, refined the writing, and reorganized several sections accordingly.
> Please kindly take a look at the updated PDF.
>
> **W1&Q1:** More Baseline
>
> **A:** Thank you for the insightful comments and for pointing out the importance of including stronger label-free baselines. We fully agree that comparing against appropriate baselines is crucial. Below we clarify why several seemingly relevant methods are not directly comparable to our setting, and what additional baselines we have added.
>
> **Regarding methods such as Self-Consistency Preference Optimization (SCPO) and TTRL:**
>  Although these works label themselves as “zero-shot,” they still rely on real task questions as inputs during optimization. Their performance is therefore substantially determined by the selection and quantity of those seed problems. Because our method does not require any real questions, labeled data, or seed prompts of any kind, these approaches are not directly comparable, and a head-to-head comparison would not be fair to either side. In other words, these methods are “label-free” but not data-free, whereas our setting is strictly data-free. Thank you for pointing out our omission of the related work, SCPO. We have now added it to the paper.
>
> **Other potential baselines:**
>  We agree that additional baselines would be helpful. However, works such as Self-Challenging Language Model Agents and Co-evolving LLM Coder and Unit Tester are specifically designed for agentic or interactive environments, not general reasoning tasks. Their training signals depend on environment feedback, code execution, or verifiable task outcomes, which again makes them fundamentally different from our purely offline, data-free setup. Therefore they cannot be directly applied to general reasoning benchmarks used in our paper.
>
> **Newly added baseline:**
> To address this concern, we added Absolute Zero as a baseline. Absolute Zero relies on a verifiable coding environment and a set of structured, executable rules. It achieves strong improvement on coding tasks (e.g., HEval) and also brings gains on math reasoning (e.g., Math500) performance. However, the requirement inherently narrows the method’s applicability to domains that can be expressed as deterministic programs.
> In contrast, our R-Zero does not rely on a Python execution environment or any form of externally verifiable supervision (so inapplicable to coding tasks), but instead employs a fully “label-free” paradigm. This frees R-Zero from the structural constraints of coding-based verification and enables much broader generalization, allowing self-evolution to operate effectively across a wide variety of mathematical and general reasoning tasks
> Through extensive comparison, we find that R-Zero outperforms Absolute Zero on mathematical and general reasoning benchmarks. We have included the complete performance results in the Table 1.
>
>
> | Model                   | Math Average | General Average |
> |-------------------------|--------------|-----------------|
> | Qwen3-4B-Base           | 42.57        | 26.34           |
> | AZR                     | 46.42        | 29.93           |
> | R-Zero                  | 49.93        | 31.15           |
> | Qwen3-8B-Base           | 48.64        | 31.98           |
> | AZR                     | 52.68        | 34.39           |
> | R-Zero                  | 53.72        | 34.50           |
> | OctoThinker-3b-base     | 26.64        | 7.47            |
> | AZR                     | 27.23        | 16.03           |
> | R-Zero                  | 29.32        | 11.12           |
> | OctoThinker-8b-base     | 36.41        | 11.70           |
> | AZR                     | 36.60        | 18.40           |
> | R-Zero                  | 38.52        | 23.00           |

---

> > ### Author Response · Authors · 2025-11-19
> > **Response to Reviewer XCxR (Part 2)**
> >
> > **W2&W6(1):** Moving Sec2 to appendix for more experiments and explaination
> >
> > **A:** Thank you for the helpful suggestion. We agree that providing a clear pseudocode description would improve the readability of the method, especially given its GAN-style iterative interaction. Following your recommendation, we have added a **step-by-step pseudocode in Appendix G**, which we believe offers an intuitive view of the full algorithmic process. We sincerely appreciate your feedback that led to this improvement.
> > Regarding the introduction of GRPO and RLVR: our questioner component crucially relies on reinforcement-learning–style optimization. Unlike supervised fine-tuning, our setting does not assume a single “correct” output for a given prompt; instead, the training signal is based on relative preference between questions. Therefore, presenting the underlying RL mechanisms helps readers understand why SFT-based approaches are insufficient and how our optimization process operates.
> > That said, we appreciate your point about space and clarity. **We have moved part of this detailed discussion to the appendix to streamline the main text while still keeping the necessary background for reproducibility.**
> >
> > **W3:** Out-of-Math training data
> >
> > **A:** Thank you for the insightful comment. To address your concern about the training data being limited to the math domain, we conducted an additional experiment where, during problem generation, we removed the requirement that the model must produce math questions. As a result, the generated training data contained a **mixture of commonsense questions and other general problem types**. The experimental results are shown below:
> > | Change Prompt               | math_average | general_average |
> > |-----------------------------|--------------|-----------------|
> > | Qwen3-4B                    | 42.57  | 21.94     |
> > | R-Zero                      | 49.93  | 31.15     |
> > | R-Zero (remove math prompt) | 49.61  | 31.07     |
> > | Qwen3-8B                    | 48.64  | 29.59     |
> > | R-Zero                      | 53.72  | 34.50     |
> > | R-Zero (remove math prompt) | 54.12  | 35.33     |
> >
> >
> > As we can see, when the base model is sufficiently capable (e.g., the 8B model), training on more general, non-math-specific data actually leads to **further improvements**. We have added these results to the **appendix H**, where you can also find detailed results for each dataset.
> >
> > **W4:** Hyperparameter selection
> >
> > **A:** Thank you for raising this important point. We agree that hyperparameter sensitivity is crucial for practical applicability, and we address each of the reviewer’s concerns below.
> > **1. Size of the candidate pool (N = 8,000).**
> > This value is not a sensitive hyperparameter. In fact, N can be set to a wide range of reasonable values. The only requirement is that each generated question is used at most once during training, ensuring that the Solver does not overfit to repeated pseudo-labels. In our preliminary experiments, the system reliably converges within approximately 15 training steps, so **any value of N sufficiently larger than the expected number of retained questions** would work. We simply chose 8,000 as a convenient setting.
> >
> > **2. Consistency filter threshold (δ = 0.25) and sample numbers (m=10).**
> >  This threshold directly follows the methodology used in prior work [1][2][3] We adopt the same criterion to ensure comparability and methodological consistency rather than tuning it to our advantage. This choice of m also **follows prior literature**[1]  which we will cite properly in the revision).
> >
> > [1]Zhang, J., & Zuo, C. (2025). Grpo-lead: A difficulty-aware reinforcement learning approach for concise mathematical reasoning in language models. arXiv preprint arXiv:2504.09696.
> >
> > [2]Li Z, Chang Y, Zhou Y, et al. Semantically-Aware Rewards for Open-Ended R1 Training in Free-Form Generation[J]. arXiv preprint arXiv:2506.15068, 2025.
> >
> > [3]Bercovich, A., Levy, I., Golan, I., Dabbah, M., El-Yaniv, R., Puny, O., ... & Chung, E. (2025). Llama-nemotron: Efficient reasoning models. arXiv preprint arXiv:2505.00949.
> >
> > **3. Clustering distance threshold (τ₍BLEU₎ = 0.5).**
> >  This is simply the default threshold of the clustering implementation provided by scikit-learn. We **did not modify** or tune this parameter; it serves only to avoid trivial duplicates within a batch.
> >
> > Overall, while several hyperparameters appear in the description of the R-Zero procedure, most are either inherited from prior work, chosen for standard methodological reasons, or set to default values. Our experiments indicate that the method is **not highly sensitive** to these parameters, and we will clarify this in the revision.

---

> > > ### Author Response · Authors · 2025-11-19
> > > **Response to Reviewer XCxR (Part 3)**
> > >
> > > **W5&W6(2):** Mix data from human and R-Zero
> > >
> > > **A:** We appreciate the comment and agree that practical impact is important when labeled data is available. In our experiments (Sec. 5.3), we evaluated the synergy with supervised data by comparing R-Zero alone, SFT alone, and a combined approach. As shown in the results above, integrating supervised data directly into R-Zero training provides gains over using either method alone, but the sequential strategy,  first R-Zero, then high-quality SFT, yields the best performance. This suggests that R-Zero can be effectively combined with existing labeled data, either by pre-alignment before SFT or by adding its outputs on top of standard training sets, showing orthogonal benefits. We have added the results in Sec. 4.3. **We also expand this section** by add the results of 8B model.
> > > | Method         | Score       |
> > > |----------------|-------------|
> > > | human          | 50.55       |
> > > | R-zero         | 49.93 |
> > > | R-zero+human   | 51.03 |
> > >
> > > **W7:** Time cost of R-Zero
> > >
> > > **A:** Regarding the cost analysis, the overhead introduced by our method is largely manageable. Operations such as clustering and computing answer consistency are performed entirely on CPU without invoking the LLM. This is also why we adopt BLEU—these CPU-only steps are extremely lightweight, taking roughly 10 seconds per training step, compared to about 30 minutes for the entire step, and can therefore be considered negligible.
> > > For data sampling prior to solver training, the required time is around 2 hours, while a full solver training iteration takes about 8–9 hours (our method is based on the standard sampling procedure of RLVR training paradigm). Thus the sampling cost is acceptable. The challenger training is more expensive—one iteration takes roughly twice the time of a solver iteration (about 15 hours). Since our method requires two layers of iteration, this does introduce additional cost; however,  as our method removes the need for expensive training data collection, we believe the remaining overhead is acceptable.
> > >
> > >
> > > **W8&W9:** Update Figure 1
> > >
> > > **A:** We have **updated both parts of Figure 1** accordingly in the revised version of the paper. Thank you very much. Your feedback has made our manuscript more **readable**.
> > >
> > > **W10:** Limitation Section
> > >
> > > **A:** While R-Zero demonstrates strong improvements in reasoning performance across multiple domains, several limitations remain. First, the core mechanism of R-Zero relies on domains where correctness can be objectively verified. The Challenger–Solver co-evolution process depends on clear, deterministic evaluation signals to produce reliable training feedback. Consequently, applying R-Zero to open-ended or subjective tasks, such as creative writing, dialogue, or preference-driven generation, remains difficult, as these tasks lack unambiguous correctness criteria. In addition, R-Zero currently employs specific labeling and verification strategies that may not generalize to all task types. Developing more robust and broadly applicable labeling mechanisms would further expand the range of domains where R-Zero can be effectively applied.
> > > Despite these limitations, we view R-Zero as a meaningful step toward creating fully self-evolving LLMs, and we hope future research will extend this paradigm to broader and more complex settings.
> > >
> > > We have **added** this section as Sec.6

---

> ### Comment · Reviewer_XCxR · 2025-11-20
>
> First, thanks to the authors for their significant effort. The authors revision has improved the work and will accordingly raise my score 4 --> 6, with confidence remaining the same.
>
> w1 & q1. okay, this is mostly reasonable. how is absolute zero implemented if their original method includes a verifiable environment?
>
> w2&w6. this has improved the quality of this work.
>
> w3. how did removing math prompts lead to higher average math score in the table you visualized? this does not match intuition.
>
> w4. this remains a weakness, especially for practitioners. it is unclear why hyperparameters set in GRPO-style RL works would be the optimal hyperparameters to this new algorithm. nevertheless, i agree that the contribution outweighs this weakness as this is meaningful first step towards the query and label-free direction.
>
>  w5. it would nice to have a more comprehensive experiment for this.
>
> w7. there are some clear computational overheads and algorithmic complexity, but this is in the right direction as a first step research
>
> w8&w9. the visualization has improved.
>
> w10. thanks.

---

> > ### Author Response · Authors · 2025-11-22
> >
> > Thank you for your response.
> >
> > **w1 & q1:** How is absolute zero implemented
> >
> >  **A:** Absolute Zero is implemented specifically for programming problems, where the method can reliably construct verifiable training data. Concretely, it uses a pipeline to generate (input, code, output) triples. Because the domain is code execution, Absolute Zero can simply run the input and code through a compiler or interpreter to check whether the generated triple is correct. Only triples whose execution matches the expected output are kept.
> > Once a verified triple is obtained, the framework applies a template to convert it into a natural-language question such as “Given the following input and code, what is the output?”. In this setup, the compiler effectively serves as a verifiable environment: both runtime errors and incorrect outputs are automatically detected.
> >
> >
> > **w3:** Why non-constratint works better
> >
> > **A:**  We only removed the math-specific constraint in the question-generation prompt; we did not remove math questions from the generated data. As a result, many math questions still appear naturally. Allowing non-math questions to be included increases the diversity of the training set, which strengthens the model’s overall reasoning ability. This broader reasoning improvement subsequently raises the model’s math scores as well. This effect is especially pronounced for larger models, which benefit more from diverse reasoning supervision.

---

> > > ### Comment · Reviewer_XCxR · 2025-11-28
> > >
> > > Thank you to the authors for their effort.
> > > I will maintain my score.

---

> ### Author Response · Authors · 2025-11-29
>
> Thank you for your reply, and increased score from 4 to 6.

---

### Official Review · Reviewer_Qxwk · 2025-10-29

**Soundness:** 4
**Presentation:** 3
**Contribution:** 3
**Rating:** 8
**Confidence:** 4

**Summary:**

The paper introduces R-Zero, a fully autonomous framework designed to enhance the reasoning capabilities of Large Language Models (LLMs) without relying on any human-curated data for alignment. The core problem it addresses is the significant bottleneck and cost associated with creating large, high-quality datasets for fine-tuning and reinforcement learning.

R-Zero's methodology is based on a co-evolutionary loop between two distinct roles, both initialized from the same base LLM :

The Challenger: This model's objective is to generate new, challenging problems that lie at the very edge of the Solver's current abilities. It is rewarded based on the Solver's uncertainty, which is measured by the consistency of its answers to a given problem.

The Solver: This model's goal is to solve the increasingly difficult tasks presented by the Challenger. It is trained on a filtered set of these problems using "pseudo-labels" that are generated by a majority vote of its own answers.

This self-contained cycle creates a dynamic, adaptive curriculum that progressively increases in difficulty, allowing the model to improve its reasoning skills without external supervision or pre-existing task datasets.

**Strengths:**

R-Zero's primary strength is its ability to create a self-improving loop for reasoning tasks without human-labeled alignment data. It successfully adapts the self-play paradigm to a domain that lacks a perfect external verifier (like a code executor or game engine), cleverly using a majority-vote mechanism to create a noisy but effective "pseudo-ground truth".

The framework demonstrates significant and consistent performance gains across different model architectures (Qwen3, OctoThinker) and scales. For instance, the Qwen3-8B-Base model achieved a +5.51 point increase on average across math reasoning benchmarks after three iterations.

The paper validates the importance of its key design choices. Ablation studies show that removing components like the repetition penalty (to ensure question diversity) or the task filtering mechanism (to control curriculum difficulty and quality) leads to a significant drop in performance.

**Weaknesses:**

The most significant limitation is that the self-improvement process is not indefinitely stable. After a few iterations, all tested models experience a "performance collapse," where their scores on benchmarks begin to decline. Larger models are more resilient and collapse later, but the eventual degradation appears inherent to the current framework.

The performance collapse is directly linked to a decline in the quality of the training data. As the Challenger generates progressively harder problems, the Solver's ability to form a reliable consensus via majority vote diminishes. The paper shows the true accuracy of these pseudo-labels systematically drops with each iteration (e.g., from 79.0% to 63.0%), introducing increasing noise into the training signal.

The framework's effectiveness has only been demonstrated in the domain of mathematics. This is a carefully chosen domain where answers are objective and can be verified through simple string matching, making the majority-vote mechanism viable. The method's applicability to more subjective, open-ended domains like creative writing, dialogue, or nuanced analysis remains a major, unaddressed challenge.

**Questions:**

Q1: The paper focuses on a self-evolving framework where the Challenger and Solver are initialized from the same base model. Have you considered an asymmetric or heterogeneous setup—for instance, using a more capable model as the Challenger and a less capable one as the Solver? It would be interesting to know if such a configuration could potentially delay the onset of performance collapse.

Q2: Regarding the training hyperparameters, the number of rollouts for the GRPO algorithm is relatively small. Could you elaborate on the motivation behind this choice? I'm curious if a larger number of rollouts was explored and whether the current setting risks insufficient exploration, which could potentially lead to a less accurate advantage signal.

Q3: The analysis of synergy with supervised data shows that using R-Zero as a pre-alignment step before Supervised Fine-Tuning (SFT) is beneficial. A compelling alternative would be to integrate the labeled data directly into the R-Zero training loop, perhaps by mixing it into the Solver's training dataset at each iteration. Have you experimented with this concurrent training approach, and do you have any insights on whether it might yield superior results compared to the sequential (SFT than R-Zero) method?

Q4: This is a minor suggestion for presentation clarity: in Figure 3, the x-axis is labeled with training steps (e.g., "Step 15," "Step 30"). For better consistency with the narrative, it might be more intuitive to label it with the corresponding iteration numbers (e.g., "Iteration 1," "Iteration 2," etc.).

---

> ### Author Response · Authors · 2025-11-19
> **Response to Reviewer Qxwk (Part 1)**
>
> **W1:** Performance collapse
>
> **A:** We agree with the reviewer that performance collapse is a real limitation of the current framework. Our goal in this work is to establish a *foundation* for self-evolving training, and we view stability as an important direction for future research. Many methods could be used to solve this problem as discussed in [1](document corpus) [2](LLM as a judge)
>
> [1] Liu B, Jin C, Kim S, Yuan W, Zhao W, Kulikov I, Li X, Sukhbaatar S, Lanchantin J, Weston J. SPICE: Self-Play In Corpus Environments Improves Reasoning. arXiv preprint arXiv:2510.24684. 2025 Oct 28.
>
> [2] Sun W, Cheng X, Fan J, Xu Y, Yu X, He S, Zhao J, Liu K. Towards Agentic Self-Learning LLMs in Search Environment. arXiv preprint arXiv:2510.14253. 2025 Oct 16.
>
> **W2:** Noisy signal
>
> **A: ** We agree that the performance collapse is indeed driven by the degradation of pseudo-label quality: as the Challenger produces harder queries, the Solver’s majority-vote consensus becomes less reliable, causing the true accuracy of pseudo-labels to drop across iterations. However, this issue is not unique to our framework—it is a shared limitation across all label-free or self-training methods. Complementary techniques such as LLM-as-a-Judge, confidence-based filtering, or external evaluators can be readily used to mitigate pseudo-label noise.
>
> **W3:** Unextendable to open-ened settiing
>
> **A:** We acknowledge this limitation. Our current study focuses on mathematics because, like most recent work[1][2] in self-training and label-free learning, we target domains with objective and verifiable answers, where consensus mechanisms such as majority vote are reliable. Extending the framework to open-ended domains—e.g., creative writing, dialogue, or creative reasoning—will likely require complementary techniques such as LLM-as-a-Judge, pairwise comparison, or other preference-based evaluators.
> This is indeed an open challenge and falls outside the current scope of our work, but our framework provides a foundation upon which such evaluation modules can be incorporated.
>
> [1]Zuo Y, Zhang K, Sheng L, Qu S, Cui G, Zhu X, Li H, Zhang Y, Long X, Hua E, Qi B. Ttrl: Test-time reinforcement learning. arXiv preprint arXiv:2504.16084. 2025 Apr 22.
>
> [2]Prasad A, Yuan W, Pang RY, Xu J, Fazel-Zarandi M, Bansal M, Sukhbaatar S, Weston J, Yu J. Self-consistency preference optimization. arXiv preprint arXiv:2411.04109. 2024 Nov 6.
>
> **Q1:** Larger Challenger
>
> **W1:** Thank you for this insightful suggestion. We conducted additional experiments using a stronger Challenger, **Qwen-3-32B-Instruct**. The results are summarized below:
>
>
> | Model | math-average | general-average |
> |-------|--------------|-----------------|
> | Qwen3-4B-Base | 42.58 | 26.34 |
> | R-Zero | 49.07 | 31.15 |
> | Stronger Challenger | 51.23 | 33.45 |
> | Qwen3-8B-Base | 49.18 | 31.98 |
> | R-Zero | 54.69 | 34.50 |
> | Stronger Challenger | 52.13 | 33.34 |
>
>
> As the results show, using a stronger Challenger helps notably when the base model is relatively weak (e.g., 4B)。 However, when the base model is larger (e.g., 8B), the stronger Challenger provides less benefit.
> A plausible explanation is that the co-evolving Challenger adapts alongside the Solver and can propose increasingly suitable and challenging problems, whereas a fixed stronger model remains static.
>
> **Q2:** Small rollouts
>
> **A:** Thank you for the question. For the Solver’s training, we follow the default hyperparameter settings used in **EasyR1**, which adopts a relatively small number of rollouts per prompt.
>
>
> For the Challenger’s training, although the number of rollouts appears small, our setup effectively provides **much larger exploration**. This is because the Challenger always receives the *same input prompt* within a batch, meaning that all rollouts in the entire batch collectively contribute to exploration. Importantly, our penalty term is also computed across the whole batch, so the effective number of rollouts is the *sum of rollouts over the entire batch*, rather than the per-prompt rollout count typically used in GRPO.

---

> ### Author Response · Authors · 2025-11-19
> **Response to Reviewer Qxwk (Part 2)**
>
> **Q3:** Mix human data and R-Zero data
>
> **A:** We evaluated the effect of integrating supervised data directly into the R-Zero training loop by mixing it into the Solver’s dataset at each iteration.This concurrent approach outperforms using either R-Zero or supervised fine-tuning (SFT) alone, but it does not surpass the sequential method of first applying R-Zero and then performing SFT. We hypothesize that mixing the labeled data during R-Zero training may dilute the high-quality signal while partially mitigating noise, whereas performing R-Zero first allows the model to acquire reasoning ability before leveraging the high-quality supervised data, which appears to be the most effective strategy. The precise underlying reasons require further investigation.  The full results are shown in sec 4.3
> | Method         | Score       |
> |----------------|-------------|
> | human          | 50.55       |
> | R-zero         | 49.93 |
> | R-zero+human   | 51.03 |
>
> **Q4:**  Presentation clarity
>
> **A:** We have updated the figure to make our manuscript more readable. Thank you for your suggestion.

---

> ### Comment · Reviewer_Qxwk · 2025-11-28
>
> I appreciate the authors' detailed response. Since the rebuttal has effectively addressed my questions, I will keep my original rating.

---

### Official Review · Reviewer_kwkW · 2025-10-31

**Soundness:** 3
**Presentation:** 3
**Contribution:** 3
**Rating:** 6
**Confidence:** 4

**Summary:**

This work proposes R-Zero, a co-evolving framework that generates new training data from scratch. R-Zero consists of two independent models, the Challenger and the Solver, which work cooperatively and are trained separately within a co-evolving pipeline. The Challenger is trained based on an uncertainty reward from the Solver model and a repetition penalty given the diversity of its generated training batch. The solver is trained with the RLVR objectives. It conducts experiments on Qwen-3 and OctoThinker, where R-Zero shows noticeable improvements over the base model without training and a fixed challenger variant in the math domain and a set of general-domain reasoning benchmarks, including SuperGPQA, MMLU-Pro, and BBEH. This work finally proposes an analysis in terms of the effects of the repetition penalty and the filtering, and eventually shows that R-Zero serves as a good mid-training strategy for enhancing model’s post-finetuning on labelled datasets.

**Strengths:**

1. Generating novel data from scratch is a valuable research direction, where the co-evolving framework that includes a dual-agent setup is novel and insightful.

2. The experiments conducted in this work are extensive, and the empirical performance improvement appears to be large.

3. The ablation study is thorough and provides fruitful findings for future research in this direction.

**Weaknesses:**

1. The baseline studied in this work is relatively weak. There are other data generation approaches, such as Absolute Zero [1], which have been discussed but not directly compared empirically.

2. Meanwhile, as the author also mentions, the RLVR methods with zero-shot training objectives, such as maximizing the model confidence and entropy, also need to be compared against the data generation approaches, given that they are all a form of zero-shot approaches.

[1] Zhao, A., Wu, Y., Yue, Y., Wu, T., Xu, Q., Yue, Y., Lin, M., Wang, S., Wu, Q., Zheng, Z., & Huang, G. (2025). Absolute Zero: Reinforced Self-play Reasoning with Zero Data (No. arXiv:2505.03335). arXiv. https://doi.org/10.48550/arXiv.2505.03335

**Questions:**

1. Could you please clarify and compare more precisely your approach with the Absolute Zero method?

2. In the R-Zero framework, you propose independent roles of the challenger and the solver. What if we train the same model with both roles?

---

> ### Author Response · Authors · 2025-11-19
> **Response to Reviewer kwkW**
>
> Thank you very much for your valuable comments. We have revised the manuscript to incorporate your suggestions, and we will address your concerns in detail below:
>
> **W1&Q1:** Comparsion to Absolute Zero
>
> **A:** We added Absolute Zero as a baseline. Absolute Zero relies on a verifiable coding environment and a set of structured, executable rules. It achieves strong improvement on coding tasks (e.g., HEval) and also brings gains on math reasoning (e.g., Math500) performance. However, the requirement inherently narrows the method’s applicability to domains that can be expressed as deterministic programs.
> In contrast, our R-Zero does not rely on a Python execution environment or any form of externally verifiable supervision (so inapplicable to coding tasks), but instead employs a fully “label-free” paradigm. This frees R-Zero from the structural constraints of coding-based verification and enables much broader generalization, allowing self-evolution to operate effectively across a wide variety of mathematical and general reasoning tasks
> Through extensive comparison, we find that R-Zero outperforms Absolute Zero on mathematical and general reasoning benchmarks. We have included the complete performance results in the updated PDF (Table 1 and Table 2).
> | Model                   | Math Average | General Average |
> |-------------------------|--------------|-----------------|
> | Qwen3-4B-Base           | 42.57        | 26.34           |
> | Absolute Zero                     | 46.42        | 29.93           |
> | R-Zero                  | 49.93        | 31.15           |
> | Qwen3-8B-Base           | 48.64        | 31.98           |
> | Absolute Zero                     | 52.68        | 34.39           |
> | R-Zero                  | 53.72        | 34.50           |
> | OctoThinker-3b-base     | 26.64        | 7.47            |
> | Absolute Zero                     | 27.23        | 16.03           |
> | R-Zero                  | 29.32        | 11.12           |
> | OctoThinker-8b-base     | 36.41        | 11.70           |
> | Absolute Zero                     | 36.60        | 18.40           |
> | R-Zero                  | 38.52        | 23.00           |
>
> **W2:** Comparision to Label-Free RLVR
>
> **A:** These methods are not directly comparable with R-Zero because they still rely on human-created problems (or say questions) – even though not require labels. In contrast, R-Zero does not use any externally provided problems or labels. In other words, these methods are **label-free but not data-free**, whereas our setting is strictly data-free.
> Their performance is therefore substantially determined by the **selection and quantity of those seed problems.** Therefore, the assumptions and data requirements of these RLVR approaches differ fundamentally from our R-Zero.
>
> **Q2:** One model for two role
>
> **A:** Thanks for your question! We have already trained the same model with both roles (referred to as Single-R-Zero), and reported in Appendix D. If the reviewers find it helpful, we would be happy to move these experiments into the main paper.
> To summarize the results (we copied the results from Appendix D to below), using two independent models is crucial for both performance and stability. Specifically, the standard two-model R-Zero framework achieves a higher peak performance (49.12) and sustains improvement for more iterations, whereas the unified Single-R-Zero model peaks at the first iteration and deteriorates immediately. Moreover, the Single-R-Zero setup consistently produces pseudo-labels of lower accuracy (e.g., 63.4% vs. 71.0% in the first iteration), likely due to internal bias and overconfidence when the same model must both generate and solve its own problems.
> These observations further justify our design choice of maintaining separate Challenger and Solver models.
>
>
> | Iteration | R-Zero Performance | R-Zero Pseudo-label Acc (%) | Single-R-Zero Performance | Single-R-Zero Pseudo-label Acc (%) |
> |-----------|---------------------|------------------------------|-----------------------------|-------------------------------------|
> | **Step 15** | 48.06 | 71.0 | **47.31** | 63.4 |
> | **Step 30** | 48.44 | 56.2 | 46.95 | 46.6 |
> | **Step 45** | **49.12** | 48.8 | 45.57 | 32.6 |
> | **Step 60** | 46.52 | 42.2 | 43.89 | 33.8 |

---

> > ### Comment · Reviewer_kwkW · 2025-11-22
> > **Response to the author**
> >
> > Thank you for the detailed response. It has sufficiently addressed my remaining concerns and I will keep my current rating.

---

### Official Review · Reviewer_Bdrg · 2025-11-01

**Soundness:** 3
**Presentation:** 3
**Contribution:** 3
**Rating:** 6
**Confidence:** 4

**Summary:**

This paper proposes R-Zero, a fully autonomous self-evolving reasoning framework that requires no human-annotated data. The framework uses a Challenger model to generate challenging tasks that lie near the capability boundary of a Solver model, which in turn learns to solve them. The system iteratively co-evolves without any external supervision. Experiments show consistent improvements in both mathematical reasoning (e.g., GSM8K, MATH, AIME) and general reasoning benchmarks (e.g., MMLU-Pro, SuperGPQA, BBEH).

**Strengths:**

The paper reports consistent performance improvements across both mathematical and general reasoning benchmarks.

It conducts rich and detailed ablation studies, revealing several interesting phenomena.

1. Models fine-tuned after R-Zero pretraining perform better than those fine-tuned directly.

2. Both task filtering and repetition penalty are shown to be essential components.

3. The paper identifies a model collapse phenomenon after multiple self-evolution iterations, with larger models showing better resistance.

4. The accuracy of self-generated pseudo-labels gradually degrades (from 79% to 63%), highlighting an important data quality issue.

5. Separating the Challenger and Solver is shown to be necessary to avoid overfitting and instability.

**Weaknesses:**

Some of the reported improvements in Table 1 appear to be statistically insignificant, weakening the empirical strength of the main claims.

**Questions:**

The authors have already provided extensive and detailed ablation studies, so I do not have additional questions.

---

> ### Author Response · Authors · 2025-11-19
> **Response to Reviewer Bdrg**
>
> Thank you very much for your valuable comments. We have revised the manuscript to incorporate your suggestions, and we will address your concerns in detail below:
>
> **W1:** Insignificant improvement
>
> **A:** Thank you for the insightful comment. We acknowledge that some individual improvements in Table 1 are not statistically significant. However, our goal is to evaluate the overall robustness of the proposed method across a diverse set of benchmarks. Therefore, we report aggregate performance over many benchmarks, which provides a more reliable and stable indication of overall effectiveness. While certain tasks show small gains, the consistent improvement in the averaged results demonstrates that our method achieves overall superiority over the baselines. We also updated the results of aime from pass@1 to mean@32 to make it more stable.
>
> In addition, it is worth emphasizing that our method does not rely on any human-annotated data. Achieving improvements under this constraint already highlights the effectiveness and practicality of our approach. Furthermore, the experiments in Sec. 3.2 show that when human-labeled data is available, our method can leverage it and achieve even stronger performance, which further corroborates the validity and general applicability of our approach.

---

> > ### Comment · Reviewer_Bdrg · 2025-11-23
> >
> > Thank you for your timely and frank response. I appreciate the clarifications and revisions you have made to address my concerns. After reviewing your reply, I will maintain my original score.

---

### Author Response · Authors · 2025-11-19
**Summary of the Update and New Revision of our Paper**

We sincerely thank all reviewers for their thoughtful feedback. We are glad that the reviewers found the problem addressed is valuable, our motivation is well-supported, our experiments are comprehensive and show clear improvements, and the paper is well-organized and easy to follow.
Based on the feedback, we have updated our manuscripts accordingly and uploaded a new version of our paper for review. The changes are colored in blue. We summarize the key changes:
- We added Absolute Zero as a new baseline in Table 1 and Table 2.
- We added a new ablation study analyzing the effect of mixing human-annotated data with self-generated data (Sec. 4.3), and we expanded this section to include the results of the 8B model.
- We added a Limitation section (Sec. 6).
- We included pseudocode to facilitate understanding of R-Zero in Appendix G.
- We added an additional analysis that removes the constraint requiring the questioner to generate only math-related questions (Appendix H).
- We moved Sec. 2 (Preliminaries) into Appendix I.
- We updated some figures to make it more readable.
- We changed the results of AIME from pass@1 to mean@32, which makes the results more stable.

We sincerely thank the reviewers again for your time and suggestions. Hope you can reconsider the assessment of our work.

---

### Meta-Review · Area_Chair_fAdX · 2026-01-10

**Summary:**

This paper introduced a data-free (no training question and label are given) RL framework for enhancing LLM’s reasoning capability. The core contribution is a novel framework that introduces two independent models, Challenger and Solver, to divide the roles into question generation and training on synthesized data. The strength of this work is showing that the reasoning capability of LLM can be self-evolved without human annotation and even the question. The extensive experimental results demonstrate the superiority of the proposed method.

**Reviewer Concerns:**

Initially, the reviewers raised some concerns on the experimental settings (e.g., weak baselines, performance collapse, compatibility with human data, math-focused training). But all of them are well resolved during the rebuttal periods, as explicitly denoted by the reviewers’ responses.

**Reviewer Scores:**

Initially, the reviewers' scores are (6,6,8,4). After successful rebuttal, the reviewer's scores are changed to (6,6,8,6). All reviewers are satisfied the responses and there is no remaining concerns that were raised by the reviewers.

---

### Decision · Program_Chairs · 2026-01-26

Accept (Poster)